# Damage Quantification and Identification in Structural Joints through Ultrasonic Guided Wave-Based Features and an Inverse Bayesian Scheme

**DOI:** 10.3390/s23084160

**Published:** 2023-04-21

**Authors:** Wen Wu, Sergio Cantero-Chinchilla, Wang-ji Yan, Manuel Chiachio Ruano, Rasa Remenyte-Prescott, Dimitrios Chronopoulos

**Affiliations:** 1Institute for Aerospace Technology, Resilience Engineering Research Group, The University of Nottingham, Nottingham NG7 2RD, UK; 2Department of Mechanical Engineering, University of Bristol, Bristol BS8 1TR, UK; 3State Key Laboratory of Internet of Things for Smart City, Department of Civil and Environmental Engineering, University of Macau, Macau 999078, China; 4Guangdong-Hong Kong-Macau Joint Laboratory for Smart Cities, University of Macau, Macau 999078, China; 5Department of Structural Mechanics and Hydraulic Engineering, Andalusian Research Institute in Data Science and Computational Intelligence (DaSCI), University of Granada (UGR), 18001 Granada, Spain; 6Resilience Engineering Research Group, Faculty of Engineering, University of Nottingham, University Park, Nottingham NG7 2RD, UK; 7Department of Mechanical Engineering & Mecha(tro)nic System Dynamics (LMSD), KU Leuven, 9000 Leuven, Belgium

**Keywords:** guided waves, joints/bounded structures, damage identification, Bayesian inference, hybrid wave and finite element, surrogate model

## Abstract

In this paper, defect detection and identification in aluminium joints is investigated based on guided wave monitoring. Guided wave testing is first performed on the selected damage feature from experiments, namely, the scattering coefficient, to prove the feasibility of damage identification. A Bayesian framework based on the selected damage feature for damage identification of three-dimensional joints of arbitrary shape and finite size is then presented. This framework accounts for both modelling and experimental uncertainties. A hybrid wave and finite element approach (WFE) is adopted to predict the scattering coefficients numerically corresponding to different size defects in joints. Moreover, the proposed approach leverages a kriging surrogate model in combination with WFE to formulate a prediction equation that links scattering coefficients to defect size. This equation replaces WFE as the forward model in probabilistic inference, resulting in a significant enhancement in computational efficiency. Finally, numerical and experimental case studies are used to validate the damage identification scheme. An investigation into how the location of sensors can impact the identified results is provided as well.

## 1. Introduction

Structural joints are essential to connect different components within any large structure. These elements typically play a vital role in the load-bearing capacity of the structure [1,2,3]. One typical application of a joint in a steel structure is shown in Figure 1. During the service stage of joints, defects caused by corrosion or fatigue can lead to catastrophic failure of the structure. Even when these defects are visible, they are not easily accessible for visual inspection [4]. Therefore, there is a compelling need for accurate and efficient detection and identification of information on the health state of such joints.

In the context of sustainable industrial development, ensuring the reliability and sustainability of structures is of paramount importance. There are numerous established techniques for structural health monitoring (SHM) and non-destructive testing (NDT) which can help to this objective. Ultrasonic guided waves comprise one of these techniques, and are currently revolutionizing the approach to NDT and SHM [5] because of their high sensitivity to minor damage and their online monitoring capabilities. Guided wave testing has been applied to various structural forms, including plates [6,7], beams [8,9], pipes [10,11], and adhesive joints [12], and several different damage assessment methods have been presented.

Recent works have contributed to damage quantification and characterization in joint structures using guided waves. Rucka [13] performed longitudinal and flexural wave propagation modelling using the spectral element method in the time domain. Wave speeds and reflection times were used for damage detection. Allen [12] investigated detection of debonding at the adhesive joint using a nonlinear Lamb wave mixing approach. Fakih [14] proposed a novel framework for damage detection, localization, and assessment using ultrasonic measurements in a dissimilar material joint. A hybrid method for damage detection and condition assessment of hinge joints in hollow slab bridges using physical models and vision-based measurements was proposed in [15]. Allen and Ng [16] proposed a method to evaluate applied torque levels in bolted joints by combining harmonics generated due to nonlinear Lamb wave mixing and contact acoustic nonlinearity. Lyathakula [17] developed an integrated damage diagnostic–prognostic framework for remaining useful life estimation in adhesively bonded joints under fatigue loading. Wu et al. [18] developed a fast inspection technique for weld defects in a steel T-welded joint structure using Rayleigh-like feature guided waves. Their method utilized the semi-Analytical Finite Element method to acquire modal solutions. Except for the first and last mentioned studies for damage detection in joints, these studies rely on traditional finite element simulations for guided wave propagation and damage interaction, which can be inefficient. Additionally, there is ample room for further research and exploration in the field of damage identification in joints.

As evidenced by the above reviewed papers, numerical models of wave propagation and wave damage interaction play an important role in damage characterization, in particular for physics-based methods [7,8,19]. WFE is one of the most popular approaches; it can fully exploit the advantage of the traditional finite element model while being even more efficient. Below, we provide a review of works that simulate joints using WFE. Renno and Mace [20] combined finite element and wave finite element methods to calculate the scattering coefficients of a joint; the numerical cases of an L-frame, lap-jointed laminated beams, and an orthotropic beam with a slot were used to illustrate their approach. Mitou [21] investigated the wave propagation and scattering coefficients of joined structures composed of one joint and different numbers of plates. Aimakov [22] proposed a semi-analytical method for computing energy scattering coefficients for joints connecting an arbitrary number of semi-infinite orthotropic plates. Denis [23] assessed the reflection and transmission coefficients of waves around defects and curved joints. An optimization procedure was proposed to magnify the amplitude of the signals reflected by defects to guide the design of curved joints. Chronopoulos [24] quantified guided wave interaction effects modelled using the WFE with localized structural nonlinearities within complex composite structures. The proposed approach enabled generation of higher harmonics and sub-harmonics through harmonic balance projection. Malik [25] proposed a WFE-based approach for complete transient simulation of ultrasonic guided waves in one-dimensional waveguides. The scattering coefficients of composite beams with three types of damage, namely, notches, cracks, and delamination, were calculated, and a model reduction strategy was adopted to reduce the calculation time. Takiuti [26] conducted an initial investigation of Lamb wave scattering from discontinuities associated with high-frequency corrosion-like damage. However, few works have considered the use of wave–damage interactions for practical applications in joints.

The calculation efficiency is one of the main bottlenecks complicating successful implementation of physics-based damage identification approaches [6]. Wu et al. [27] developed a dedicated physics-based Bayesian framework for extracting damage characteristics from ultrasound measurements in plate-like structures. A semi-analytical forward model was employed to perform rapid computations of wave–damage interactions, improving the robustness and efficiency of the inversion procedure. Fakih [14] used an artificial neural network-based surrogate model with Approximate Bayesian Computation for increased computational efficiency. However, analytical solutions have restricted application, and it is not always appropriate to increase efficiency by including them. Additionally, the latter study’s method employed a neural network, which often necessitates additional sample data [28].

The goal of the current study is to create a novel method for dealing with damage identification goals in three-dimensional joints with any shape and finite size. To this end, a method for identifying the size of circular holes in a joint formed in an aluminium plate is proposed using the Bayesian inverse procedure. The proposed Bayesian framework uses a kriging-based surrogate model of the WFE approach to obtain the scattering coefficients corresponding to different defect sizes numerically. The continuity and equilibrium conditions of the joint with respect to each waveguide are used to solve the scattering feature. This study uses a particular joint form as a case study. However, using WFE it is simple to extend it to any shape. The referenced surrogate model is trained on a database containing measured scattering properties to alleviate the computational burden. Furthermore, considering that the geometry of the joint is relatively small, the scattered signal caused by the defect is difficult to distinguish from the signal reflected from the boundary; thus, a clear scattered signal cannot be obtained. Therefore, the steady-state waveform is chosen to excite the joint in the monitoring test. Then, a damage feature in the frequency domain is obtained using a specific signal processing method, by which scattering coefficients are extracted from the time-domain experimental signals. Finally, the proposed method is validated through a full finite element simulation and a physical experimental case. Finally, the consequences of different sensor locations are assessed and discussed.

The rest of this manuscript is organized as follows. Section 2 introduces the experimental study on plate joints using guided wave monitoring tests. Section 3 outlines the physics-based Bayesian inference framework used to identify the amount of damage, including the formulation of the wave finite element model described in Section 3.2 and the kriging surrogate model described in Section 3.3. In Section 4 and Section 5, numerical and experimental cases are respectively provided. Finally, our conclusions are presented in Section 6.

## 2. Guided Wave Monitoring Testing of Joints and Damage Feature Extraction

This section explores the feasibility of damage identification in aluminium joints through experimental studies. Defect-related scattering coefficients are extracted and assessed. To highlight the universality of the damage identification framework, an arbitrarily-shaped joint, shown in Figure 2, is used. The joint is based on a central plate attached to four rectangles to represent the braces of the joint, with the elements cut using a water jet for geometrical accuracy. The material properties of the aluminium plate are shown in Table 1.

Piezoelectric (PZT) sensors are placed where maximum damage sensitivity is achieved, i.e., on different braces next to the edge of the central plate. The signals and associated damage-sensitive features extracted from these data are expected to change in a monotonic fashion with increasing damage levels [29]. The position of different sensors is shown in Figure 3. The sensors are 7 mm in diameter and 0.2 mm in thickness with radial mode vibration and a resonant frequency of 300 kHz. Sensor #1 is used to generate an input waveform, and the rest of the PZT sensors receive the reflected and scattered signals. The ends of the braces are covered by plasticine to effectively reduce the influence of the reflections and create a pseudo-absorbing boundary condition. Thus, the performance of the signals scattered by the defect is enhanced.

The overall experimental setup is shown in Figure 4. A Keysight 33512B arbitrary waveform generator was used to generate a steady-state sinusoidal waveform in a specific frequency and a DSOX2014A oscilloscope was used to digitize the signals using a sampling frequency of 9.6 MHz, with an average of 32 measurements to increase the signal-to-noise ratio.

The time domain signals at 240 kHz for pristine and different damage states are shown in Figure 5. Note that after 0.25 ms the signal amplitude stabilizes, which is because the steady-state output is obtained when the steady-state waveform is excited in the linear system. After the defect is introduced, the amplitude becomes larger after stabilization. The larger the defect is, the smaller the observed amplitude of scattered waves. In this context, the scattering properties of defects are proposed for use as damage indicators. In this work, the frequency domain technique for calculation of the scattering coefficients [30,31,32] is adopted, which requires the three steps schematically illustrated in Figure 6. First, the fast Fourier transform of the incident wave is computed. Second, the incident wave is subtracted from the wave of the different damage states to obtain the scattered wave and the fast Fourier transform is computed. Finally, the coefficients are computed by dividing the frequency spectra of the reflected/transmitted part of the signal by that of the incident part.

## 3. Method

### 3.1. Outline of Bayesian Inference

Based on the damage feature analysis in Section 2, scattered fields between waves and damage interaction depend on the size of the defect. In this section, a Bayesian inference framework is presented to infer the sizes of defects based on their scattering coefficients along with quantification of the uncertainty.

The framework is based on Bayes’ theorem, as shown below [33]: (1)$$p\left(\lambda |D,M\right)=\frac{p\left(D|\lambda ,M\right)p\left(\lambda |M\right)}{p(D|M)}$$
where p(D|λ,M) is the likelihood function, which provides a measure of the agreement between the available measurement data and the corresponding numerical model output. The denominator p(D|M) is known as the evidence, and is a measure of how well the model explains the data D. It acts as a normalization constant in Bayes’ theorem [34]. The prior PDF p(λ|M) is the state of knowledge before any measurement is available. The posterior probability p(λ|D,M) is the state of knowledge of the distribution of the model parameters after updating the prior information with the measurement data. The data D can be obtained by numerical (e.g., FEM) or experimental methods (e.g., using PZT transducers, a signal generator, and an oscilloscope). M is the model class, which specifies an input/output model. The error term *e* is used to define the probabilistic damage interaction model, as follows [35,36]: (2)$$s_{D}=s_{M}\left(\lambda \right)+e.$$
where $s_{M}\left(\lambda \right)$ describes the modelled scattering coefficients obtained from the deterministic physical model and $s_{D}$ is the scattering coefficients obtained by processing the data D according to the signal processing techniques in Section 2. The deterministic physical model for determining $s_{M}\left(\lambda \right)$ is described in the following section. To apply the above theorem, a set of unknown model parameters λ are used, which include the radius of defects (*r*) and the error (e) between the modelled scattering coefficients and the scattering coefficients derived from experiments. A zero-mean Gaussian distribution with covariance matrix $\Sigma_{e}=\mathrm{diag}\left(\sigma_{e,1}^{2},\sigma_{e,1}^{2},\cdots ,\sigma_{e,N_{s}}^{2}\right)$ is selected to model the error term in order to produce the largest prediction uncertainty, i.e., $e∼N(0,\Sigma_{e})$, while $N_{s}$ refers to the dimension of the scattering coefficients based on the principle of Maximum Information Entropy [37,38]. The stochastic version of the model is provided by a Gaussian distribution
(3)$$p\left(s_{D}|s_{M},\lambda ,M\right)={\left(2\pi \sigma_{e}^{2}\right)}^{-\frac{N_{s}}{2}}\exp\left(-\frac{1}{2}{\left(\frac{J(\lambda ,D)}{\sigma_{e}}\right)}^{2}\right),$$
where J(λ,D) is a goodness-of-fit function selected to be the $L_{2}$-norm of the experimental and modelled data, defined as
(4)$$J\left(\lambda ,D\right)={\left(\sum_{i=1}^{N_{s}}{\left(s_{M,i}-s_{D,i}\right)}^{2}\right)}^{1/2}$$
with $s_{M,i}$ and $s_{D,i}$ being the *i*th element of the vectors $s_{M}$ and $s_{D}$, respectively.

The evaluation of Equation (Equation 1) involves calculating multi-dimensional integrals, for which Markov Chain Monte Carlo (MCMC) methods are widely used to estimate the posterior probability density function (PDF). MCMC methods enable direct sampling from the posterior distribution while bypassing computation of the evidence. Among the many MCMC algorithms available in the literature, the Metropolis–Hastings (M-H) algorithm [39,40] is employed here as a stochastic simulation method due to its versatility and ease of implementation. The MH algorithm is capable of avoiding calculation of the evidence, which allows Equation (Equation 1) to be rewritten as follows: (5)$$p\left(\lambda |D,M\right)\propto \left\{\prod_{k=1}^{N}p\left(D^{\left(k\right)}|\lambda ,M\right)\right\}p\left(\lambda |M\right)$$

The MH algorithm involves a random walk process that selects samples according to certain selection criteria during the sampling process. Appendix A provides more details on the implementation of the algorithm.

### 3.2. Numerical Calculation of Ultrasound Scattering for Systems Involving Beam Connections and Solid Joints of Arbitrary Complexity

A numerical approach to deriving the scattering features is presented in this section. Although traditional finite element methods for solving time-domain ultrasonic signals have proven useful, the models require large geometry to prevent unwanted reflections from boundaries, which is time-consuming. In addition to the fact that multiple modes must propagate simultaneously, this approach can lead to a substantial increase in the required geometric size of the model, especially in the presence of scatterers [41,42]. The WFE method is employed here to derive the scattering coefficients in the frequency domain, which is more computationally efficient and avoids the need for extensive signal processing. The WFE method is a technique for studying wave motion in periodic structures, for example, a short section of a waveguide or a small segment of a 2D structure [43]. The equation of motion for time-harmonic motion is obtained from a full FE model in terms of a discrete number of nodal degrees of freedom (DoFs) and forces in the same form as the dynamic stiffness method. Periodicity conditions are then applied and an eigenvalue problem is formulated, the solutions of which provide the dispersion curves and wave modes. The waveguides are modelled using the wave finite element method, and the joint is modelled using the standard FE. The DoFs at the interfaces of the waveguides and the joint are compatible. The continuity and equilibrium conditions can be used to yield the scattering coefficients [20,21,32].

#### 3.2.1. Wave Propagation in Beam Connections

If the structure undergoes time-harmonic motion at frequency ω, and in the absence of external forces, the nodal displacements and forces are related through the frequency dependent dynamic stiffness matrix of the segment as follows [44]: (6)$$\left[K+i\omega C-\omega^{2}M\right]q=f$$
where K, C, and M are the stiffness, viscous damping, and mass matrices, respectively, q denotes the displacement, and f denotes the forcing vectors. The dynamic stiffness matrix can be rearranged based on its left and right side as follows: (7)$$\left[\begin{array}{cc} D_{LL} & D_{LR}\\ D_{RL} & D_{RR} \end{array}\right]\left\{\begin{array}{c} q_{L}\\ q_{R} \end{array}\right\}=\left\{\begin{array}{c} f_{L}\\ f_{R} \end{array}\right\}$$
where the subscripts *L* and *R* denote the left and right sides of the segment, respectively. If there are non-interface nodes in the waveguides, dynamic condensation needs to be used [20,21]. Using Equation (Equation 7) and continuity of displacements and equilibrium of forces at the cross-section between sections, the following eigenvalue problem can be formulated: (8)$$\lambda \left\{\begin{array}{c} q_{L}\\ f_{L} \end{array}\right\}=T\left\{\begin{array}{c} q_{L}\\ f_{L} \end{array}\right\}$$

The transfer matrix T consists of block matrices of the dynamic stiffness matrix. The solution of the eigenvalue problem yields the wavenumber and wavemode shapes. The positive and negation-going waves can be separated, then the right and left eigenvectors grouped as in Equation (Equation 9):(9)$$\Phi =\left[\begin{array}{cc} \Phi^{+} & \Phi^{-} \end{array}\right],\;\Psi =\left[\begin{array}{c} \Psi^{+}\\ \Psi^{-} \end{array}\right]$$

The wave modes can then be normalized based on the orthogonality of the left and right eigenvectors. Finally, the displacement and forcing vectors can be expressed as
(10)$$q_{L}=\Phi_{q}^{+}a^{+}+\Phi_{q}^{-}a^{-};\;f_{L}=\Phi_{f}^{+}a^{+}+\Phi_{f}^{-}a^{-}.$$

#### 3.2.2. Calculation of Scattering Coefficients of Arbitrary Joints

Next, the calculation of scattering coefficients of arbitrary joints is illustrated. Different wave modes propagate through the waveguides, meaning that the waveguides can be considered as actuators and sensors. The time-harmonic behaviour of the joint is described through the following equation:(11)$$\left[\begin{array}{cc} {\tilde{D}}_{ii} & {\tilde{D}}_{in}\\ {\tilde{D}}_{ni} & {\tilde{D}}_{nn} \end{array}\right]\left\{\begin{array}{c} Q_{i}\\ Q_{n} \end{array}\right\}=\left\{\begin{array}{c} F_{i}\\ F_{n} \end{array}\right\}$$
where Q and F are the vectors of the DoFs and internal nodal forces, respectively, represented in the global coordinate system, and the subscripts *i* and *n* respectively represent interface and non-interface nodes. If there are no external loads, Equation (Equation 11) can be condensed as follows:(12)$$D_{ii}Q_{i}=F_{i},\;D_{ii}={\tilde{D}}_{ii}-{\tilde{D}}_{in}{\tilde{D}}_{nn}^{-1}{\tilde{D}}_{ni}\;\;\mathrm{and}\;\;Q_{n}=-{\tilde{D}}_{nn}^{-1}{\tilde{D}}_{ni}Q_{i}$$

Based on the continuity and equilibrium conditions for the joint with respect to each waveguide, we obtain
(13)$$Q_{i}=\mathrm{Rq},\;F_{i}-Rf=0$$

The vectors q and f are derived in the last section; thus,
(14)$$\left[R\Phi_{f}^{+}-D_{ii}R\Phi_{q}^{+}\right]a^{+}+\left[R\Phi_{f}^{-}-D_{ii}R\Phi_{q}^{-}\right]a^{-}=0$$

Finally, the scattering matrix can be obtained as
(15)$$s=-{\left[R\Phi_{f}^{-}-D_{ii}R\Phi_{q}^{-}\right]}^{-1}\left[R\Phi_{f}^{+}-D_{ii}R\Phi_{q}^{+}\right].$$

### 3.3. Kriging Surrogate Model with WFE

The Bayesian identification framework requires evaluation of the forward model thousands of times in order to obtain scattering coefficients corresponding to each candidate damage size. This creates a huge computational burden. Therefore, a surrogate model is used to replace the WFE forward model in order to overcome these computational challenges.

Surrogate models approximate a function based on a set of training points and then predict the function at new points. In this study, the surrogate model is used to establish the mapping relationship between the size of the damage defects and the scattered fields. Due to its fast training speed and accurate training results, a kriging surrogate model is used to map the input and outputs of the WFE model presented above. A kriging predictor is denoted by the following with an initial Design Of Experiments generated by the Latin Hypercube Design, denoted by $R={\left\{r^{\left(1\right)},r^{\left(2\right)}\cdots r^{\left(n_{s}\right)}\right\}}^{T}$, and the predicted scattering coefficients at each sample point based on the WFE model [32,45,46]:(16)$$\eta \left(r^{*}\right)=m\left(r^{*}\right)+\chi \left(r^{*}\right)$$
where $r^{*}$ is the input vector, which in this study represents the geometrical parameters of the damage; $m(r^{*})$ denotes the mean function (polynomial in $r^{*}$), an optional regression model estimated from available data; and $\chi (r^{*})$ is a Gaussian process with zero mean and the covariance function
(17)$$\mathrm{Cov}\left(r_{i},r_{j}\right)=\sigma^{2}\mathrm{Corr}\left(r_{i},r_{j}\right),\;i,j=1\ldots n_{s}$$
where $n_{s}$ denotes the number of sampled points. In the last equation, Corr represents a correlation function and $\sigma^{2}$ represents the process variance. A classical choice for this correlation function is the exponential function provided by
(18)$$\mathrm{Corr}\left(r^{\left(i\right)},r^{\left(j\right)}\right)=\prod_{k=1}^{n_{p}}\exp\left(-\vartheta_{k}{\left|r_{k}^{\left(i\right)}-r_{k}^{\left(j\right)}\right|}^{\delta }\right),\;0<\delta \leq 2$$
where $n_{p}$ denotes the number of damage characterization parameters to be inferred and $\vartheta_{k}$ denotes scale factors that can be estimated using maximum likelihood. After building the kriging surrogate model, the WFE model is replaced in the computation of the model output by
(19)$$s_{M}\left(r\right)=\eta \left(r^{*}\right)$$

It is worth noting here that all scattering coefficients are frequency-dependent. The scattering coefficients from 230 kHz to 270 kHz predicted by the surrogate model are compared with their counterparts obtained by the WFE model in Figure 7. The results show that the kriging surrogate model can predict the scattering coefficients accurately. Note that the errors induced by the surrogate model are subsumed within the error term of the Bayesian inference equation (refer to Equation (Equation 2)).

### 3.4. Workflow of the Proposed Framework

The step-by-step workflow of the proposed Bayesian framework is shown below, and is summarized in Figure 8.

Obtain the scattering coefficients $s_{D}$ for joints from ultrasonic guided wave measurements or FE model (according to the signal processing procedure provided in Figure 6).Construct a kriging surrogate model to establish the relationship between the scattering coefficients and the damage geometry information *r* using the hybrid wave and finite model introduced in Section 3.2.Formulate the likelihood function by introducing an error term that measures the difference between the modelled and experimental ultrasound data according to Equations (Equation 2), (Equation 3) and (Equation 19).Approximate the posterior distribution of the model parameters using the MH algorithm.

## 4. Numerical Validation

A full FE model of the joint shown in Figure 2 is presented here to validate the damage identification framework. The ultrasonic signal D is generated using Abaqus FEM, whereby the scattering coefficients are obtained. Figure 9 shows the FE geometry configuration. The centre of a through-thickness circular hole with radius 5 mm is located 35 mm to the left of the joint and 15 mm to the lower side. An incident steady wave is generated by applying in-plane displacement to the centre of the actuator, with a forcing function based on a steady-state sinusoidal waveform with different frequencies. The model is meshed by using 8-node general purpose linear brick elements (C3D8R) [47], with reduced integration and a maximum element edge length of 0.3 mm. The in-plane displacements are monitored at the centre of the sensors. Then, following the signal processing procedure presented in Section 2, the monitored displacements are processed to obtain the scattering field.

Samples from the posterior PDFs of model parameters are obtained through the MH algorithm (refer to Appendix A) using 100,000 samples and a Gaussian proposal distribution. The burn-in period is specified as 20,000. A uniform prior distribution was used with bounds [2.125 mm, 2.875 mm] for radius *r* and [$1\times 10^{-4}$, $1\times 10^{-2}$] for standard deviation of the error term $\sigma_{e}$. The signals from different locations were used simultaneously. The inferred results, including the maximum a posteriori values (MAP), mean, standard deviation (Std), and coefficients of variation (COV) [37], are described in Table 2. The COV is a measure of the dispersion of a variable, and is defined as the variable’s standard deviation divided by the mean. The posterior distribution of identified parameters (the radius and standard deviation) are shown in Figure 10. In terms of the MAP, the error of the radius is 2.4%, which shows a remarkable agreement between the real and inferred defect sizes.

It should be noted here that the signals from sensors at different locations can be used for damage characterization. Three different locations are investigated here, with the inferred results shown in Table 3. The sequence of the sensors is the same as that shown in Figure 9. Note that the error of the inference results varies with the position from which the signal is emitted, and is minimal when signals from S1, S2, and S3 are used simultaneously. These results demonstrate that a single SHM configuration based on a sensor–actuator pair is sufficient to identify damage; however, the inference error is reduced when signals from different locations are used simultaneously.

## 5. Validation against Physical Experiments

In this section, a case study involving a physical experimental is presented to verify the proposed framework. The experimental equipment, monitoring methods, and specimen size are exactly the same as those explained in Section 2. After obtaining the time domain signal, the frequency domain damage features are determined using the procedure described in Figure 6. A uniform prior distribution is used with bounds [1.2 mm, 3.8 mm] for the radius *r* and [$1\times 10^{-2}$, $1\times 10^{-1}$] for the standard deviation $\sigma_{e}$. Similar to the previous case study, the inferred process is carried out using the MH algorithm with 10,000 samples, and signals from different locations are used simultaneously. The inferred results, including the MAP, mean, standard deviation, and COV of the model parameters, are shown in Table 4. The posterior distribution and contour plots of the inferred parameters are shown in Figure 11. Note that the error of the inferred radius of the damage in terms of MAP is 22.0%. This relatively large error is mainly caused by the inconsistency between the ideal structural model in the finite element model and the actual specimen in the experiment, as well as to the differences between the crack shape and the actual model.

Similarly, three different locations are compared here, with the results shown in Table 5. Unlike from the inference based on the FE model, the inversion error based on S1 is the smallest. Again, simultaneous use of signals from three locations improves inversion accuracy.

## 6. Conclusions

In this paper, a Bayesian inference framework is presented for identifying the size of circular holes in joints using frequency-domain damage features. It is demonstrated that the guided wave monitoring technique is able to detect a circular hole in plate joints by exciting the steady-state waveform. A WFE model is presented to obtain the damage features numerically. To leverage the computational strategy, a kriging surrogate model is used within the Bayesian inversion scheme. Numerical and experimental studies are conducted to validate the proposed damage identification framework. In addition, the inferred accuracy as a function of sensor location is studied, finding that differences in sensor locations introduce uncertainties of varying magnitudes. In physical experiments, a pseudo-absorbing boundary condition was used to reduce the impact of boundary reflection waves on the scattering coefficient. The proposed physical model in this study theoretically avoids the influence of boundary reflection. However, when using this model, care should be taken to minimize the effect of boundary reflections on the results. Furthermore, the WFE model employed in this study cannot handle joints with bolts or other discontinuous interfaces, as these may introduce complex scattering behaviour [26]. The following conclusions can be drawn:The proposed framework provides a viable approach for damage characterization of bounded structures;The kriging surrogate model greatly improves the computational efficiency of the inversion process;The inversion error varies depending on the signal source.

Future research perspectives to be considered might include: (1) combining a neural network model with the proposed WFE model to reduce the error of the simulated experimental results and (2) investigating the effect of the frequency range on the inference error. However, selecting a suitable frequency range is beyond the scope of the current subject. We intend to investigate this topic further in our future work.

## Figures and Tables

**Figure 1 sensors-23-04160-f001:**
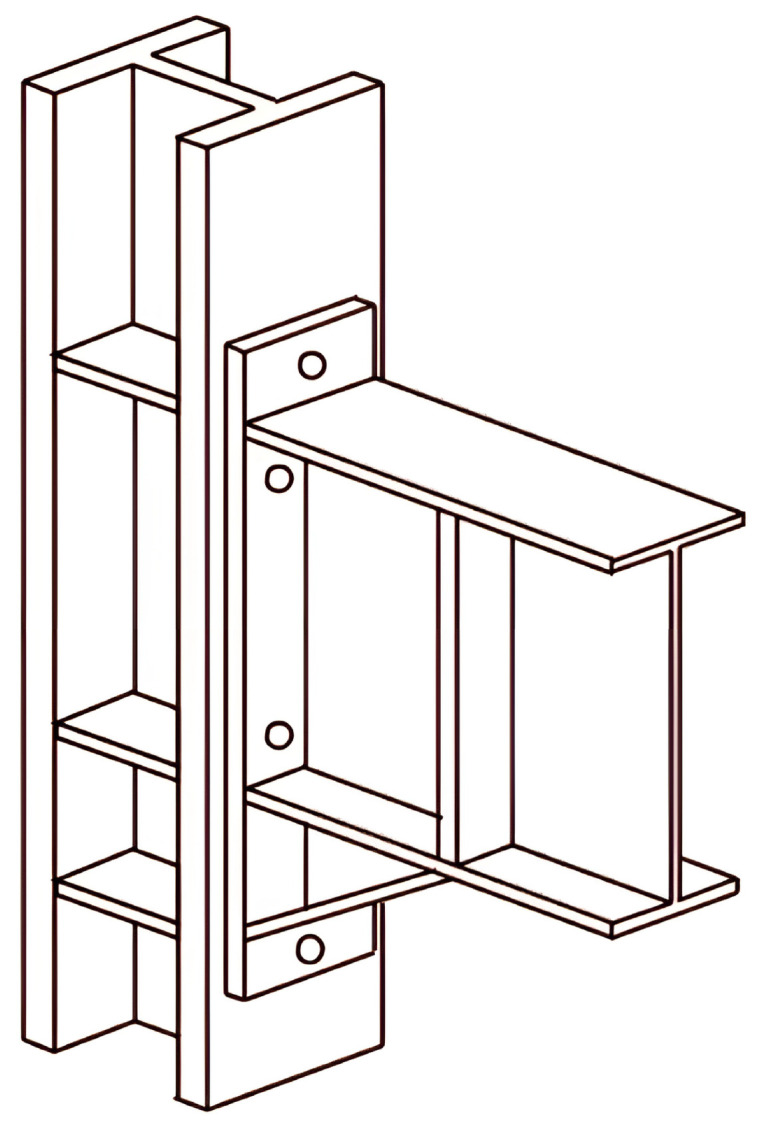
Illustrative example of joints in a steel structure.

**Figure 2 sensors-23-04160-f002:**
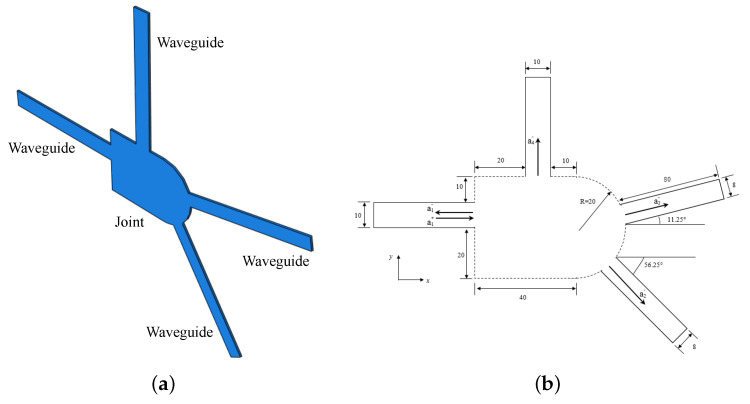
Panel (**a**): Schematic view of the aluminium joint. (**b**): detailed geometry of the joint, including its braces.

**Figure 3 sensors-23-04160-f003:**
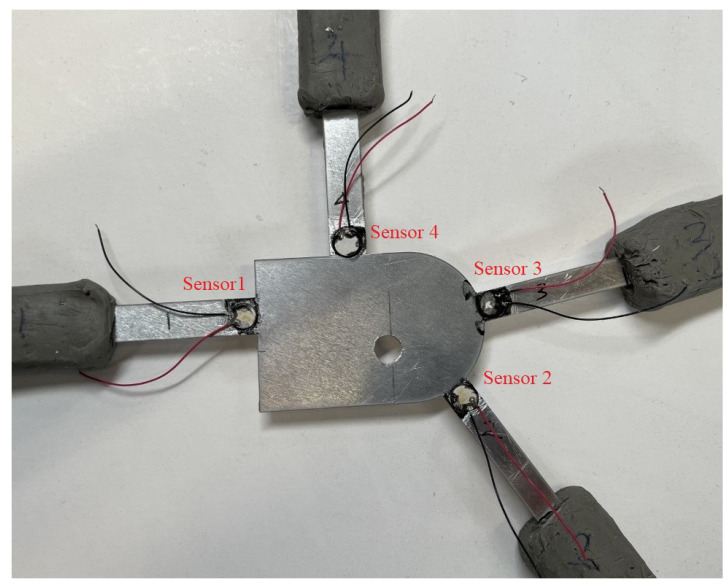
The aluminium joint with surface mount transducers and damage; there are four transducers in total, with the No. 1 transducer used to excite the signal, that is Sensor 1. The other sensors are used to receive the signal, named Sensor 2, 3 and 4 respectively.

**Figure 4 sensors-23-04160-f004:**
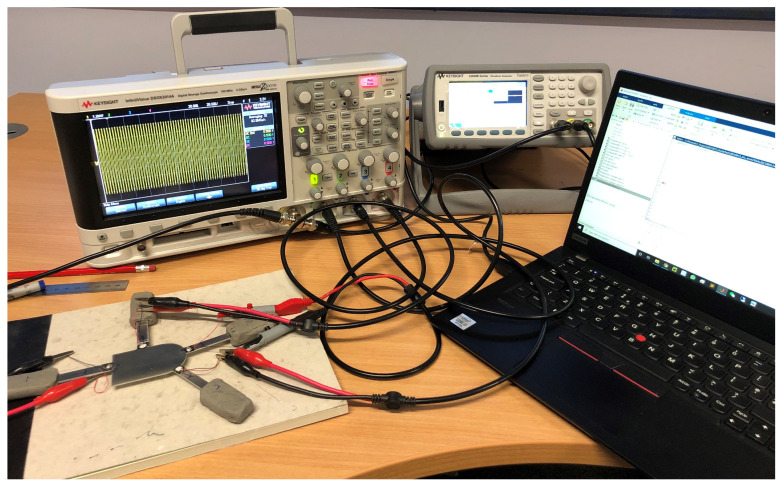
Experimental setup, comprising a laptop, an arbitrary waveform generator, and an oscilloscope connected to the PZT transducers, attached in turn to the aluminium joint.

**Figure 5 sensors-23-04160-f005:**
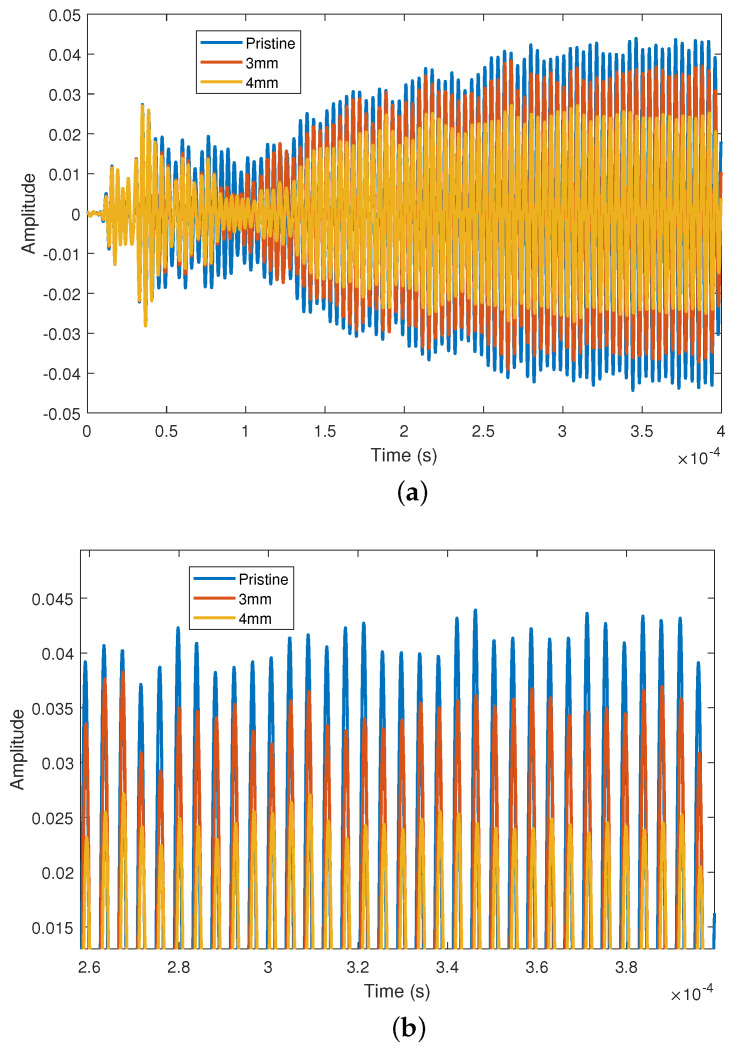
(**a**) Time domain experimental signals of different damage states at 240 kHz; (**b**) zoomed—in portion of (**a**).

**Figure 6 sensors-23-04160-f006:**
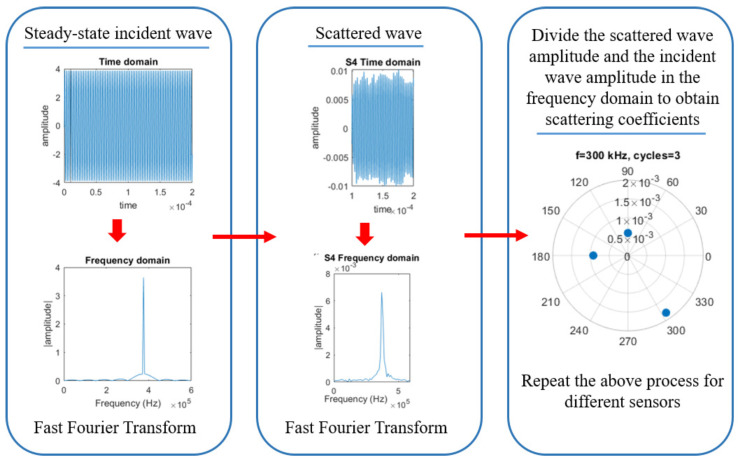
Workflow of signal processing technique used to obtain damage features in the frequency domain, namely, scattering coefficients.

**Figure 7 sensors-23-04160-f007:**
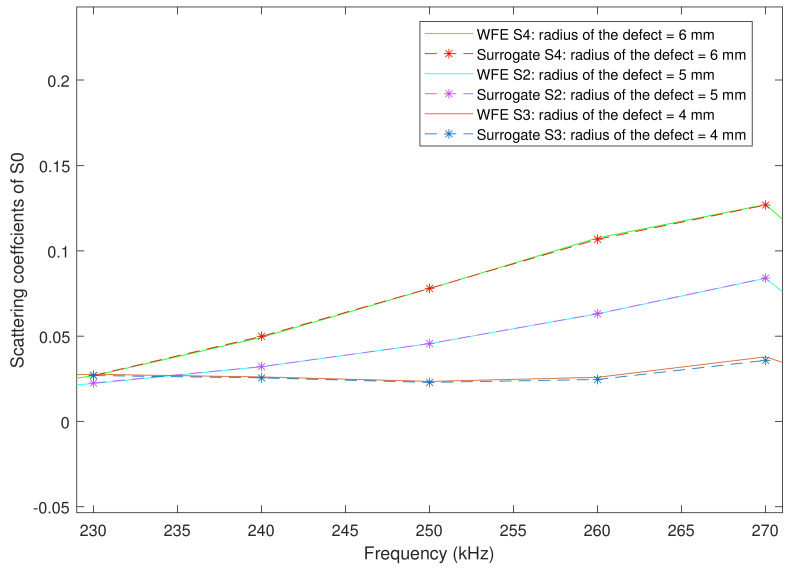
Comparison of scattering coefficients obtained by the surrogate model and WFE model; the dotted line represents the results of WFE model, the solid line represents the results of the surrogate model, and S2 and S4 represent the different locations of sensors.

**Figure 8 sensors-23-04160-f008:**
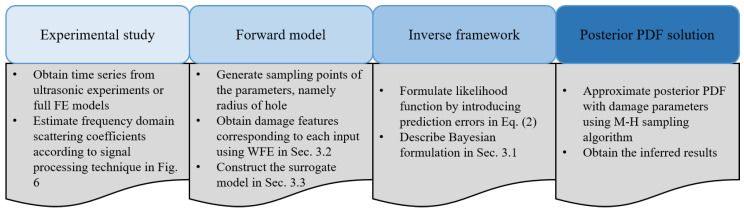
Workflow of Bayesian inference for damage identification with scattering coefficients obtained from the WFE—assisted surrogate model.

**Figure 9 sensors-23-04160-f009:**
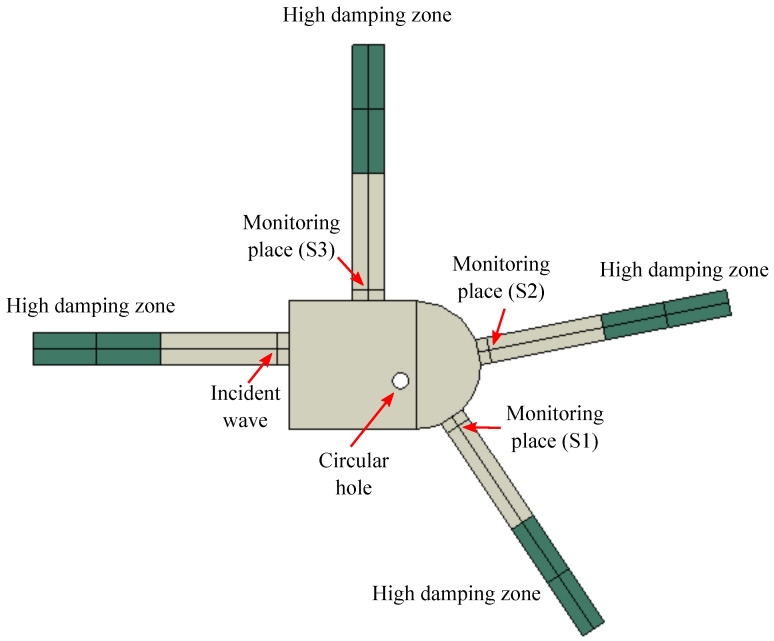
Schematic of the full finite element model for calculating scattering coefficients.

**Figure 10 sensors-23-04160-f010:**
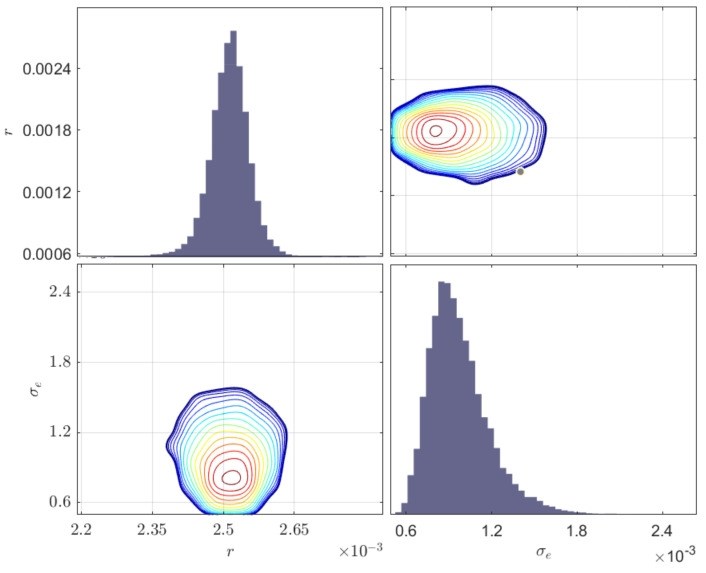
Posterior distribution of different parameters and contours of two-dimensional simulation densities inferred from full finite element model; the diagonal plots indicate the marginal distributions of the inferred parameters.

**Figure 11 sensors-23-04160-f011:**
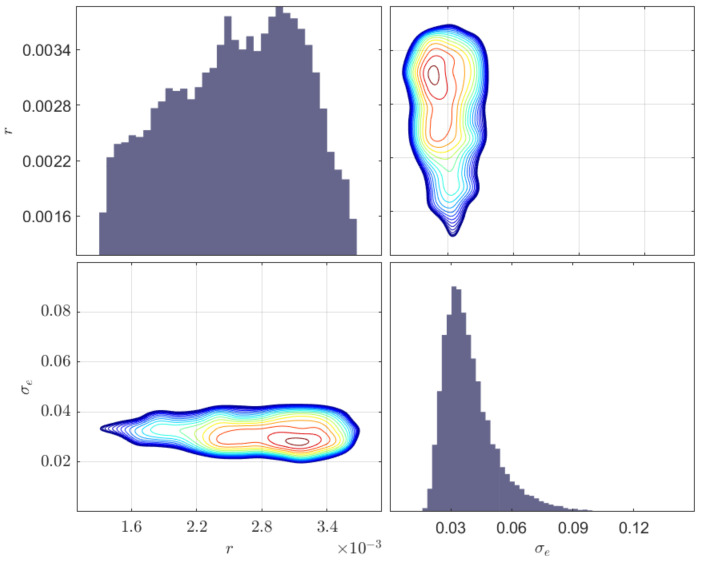
Posterior distribution of different parameters and contours of two-dimensional simulation densities inferred from physical experiments; the diagonal plots indicate the marginal distributions of the inferred parameters.

**Table 1 sensors-23-04160-t001:** Mechanical properties of aluminium plate.

Thickness (mm)	Young’s Modulus (GPa)	Poisson’s Ratio	Density (kg/m${}^{3}$)
1.2	69	0.33	2705

**Table 2 sensors-23-04160-t002:** Identified results of the numerical case from full FE.

Parameters	True Value	MAP	Mean	Std	COV (%)
*r* (mm)	2.5	2.56	2.5146	$4.9638\times 10^{-5}$	$2.464\times 10^{-9}$
$\sigma_{e}$	-	$8.256\times 10^{-4}$	$9.8091\times 10^{-4}$	$2.1799\times 10^{-4}$	$4.7519\times 10^{-8}$

**Table 3 sensors-23-04160-t003:** Comparison of results obtained by inversion of sensor signals at different positions under the numerical model.

Parameters	S1	S2	S3	S1 & S2 & S3
Errors in terms of MAP (%)	10.36	−3.74	10.2	2.4

**Table 4 sensors-23-04160-t004:** Identified results from the ultrasonic experiments.

Parameters	True Value	MAP	Mean	Std	COV (%)
*r* (mm)	2.5	3.05	2.5540	$7.0795\times 10^{-4}$	$5.0119\times 10^{-7}$
$\sigma_{e}$	-	0.031	0.0391	0.0127	$1.6247\times 10^{-4}$

**Table 5 sensors-23-04160-t005:** Comparison of results obtained by inversion of sensor signals at different positions under the physical experiment.

Parameters	S1	S2	S3	S1 & S2 & S3
Errors in terms of MAP (%)	−17.2	26.0	40.0	22.0

## Data Availability

The raw/processed data required to reproduce these findings cannot be shared at this time as the data form part of an ongoing study. They will be made available upon request.

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
