# Peer review of "Damage Quantification and Identification in Structural Joints through Ultrasonic Guided Wave-Based Features and an Inverse Bayesian Scheme"

_sensors, 2023, doi:10.3390/s23084160_

Round 1

Reviewer 1 Report

The authors describe a method to damage identification in structures subject to degradation, specifically welded joints.

The approach considers a Bayesian scheme for model updating, enabled by a surrogate model linking damage states and the observed features from a guided wave monitoring system.

The manuscript is worth of publication but requires some improvement/modification according to the following comments:

·       The authors experimentally created pseudo-absorbing boundary condition. Some comments in the conclusion regarding the applicability of the approach when this is not feasible to be implemented are needed.

·       Figure 5: the authors mention 240kHz but 300kHz was mentioned before in the text.

·       Regarding the Bayesian approach, some comments on prior selection are needed.

·       For surrogate modelling, do we really need a GPR for approximating that function? Maybe even a simpler regressor might be enough for the application.

·       Concerning the GPR, why limiting to the usage of the mean value? It is interesting to see the confidence boundaries too. Should the confidence of the GP be used as uncertainty hyperparameter of the likelihood function in the Bayes formulation?

·       Metropolis Hastings algorithm is first introduced in the Figure 8. Apparently no introduction is present in the manuscript, while the authors say in Section 5 “Similarily to the previous case study, the inferred processi s carried out using the M-H algorithm”

·       Review of typos is needed

Author Response

The authors would like to thank the reviewer for her/his constructive comments. We hope that we have adequately addressed the comments below and we are looking forward to receiving your response on our manuscript. 

Reviewer #1: The authors describe a method to damage identification in structures subject to degradation, specifically welded joints. The approach considers a Bayesian scheme for model updating, enabled by a surrogate model linking damage states and the observed features from a guided wave monitoring system. The manuscript is worth of publication but requires some improvement or modification according to the following comments:

1) The authors experimentally created pseudo-absorbing boundary condition. Some comments in the conclusion regarding the applicability of the approach when this is not feasible to be implemented are needed.

Response: The authors thank the reviewer for this valuable comment. The proposed identification framework does have this limitation. Further explanation has been added. Please check the details in lines 292-298 of the revised manuscript.

2) Figure 5: the authors mention 240kHz but 300kHz was mentioned before in the text.

Response: The authors mention 300 kHz twice in the manuscript. The first is to state that the resonant frequency of the actuator is 300 kHz. The second is a reference to Figure 5, and the second is a typo because Figure 5 shows a time-domain signal at 240 kHz.

3) Regarding the Bayesian approach, some comments on prior selection are needed.

Response: The prior information is about the uncertain model parameter λ, which is r (radius of the defect) before any model updating is performed and that this knowledge may come from engineering experience, material information, etc. In this study, the prior information is uninformative, because we only know what the upper and lower bounds of the expected defect are. So uniform prior distributions with bounds of the model parameters are given.

4) For surrogate modelling, do we really need a GPR for approximating that function? Maybe even a simpler regressor might be enough for the application.

Response: Firstly, linear regression is unable to deal with this problem, as the scattering coefficients not only depend on the defect size, but also frequency. Secondly, as the frequency increases, the relationship between frequency and scattering coefficient becomes highly nonlinear. As a result, the kriging model is the best approach to approximating the scattering coefficients [R1].

5) Concerning the GPR, why limiting to the usage of the mean value? It is interesting to see the confidence boundaries too. Should the confidence of the GP be used as uncertainty hyperparameter of the likelihood function in the Bayes formulation?

Response: In this study, we accounted for the error in the scattering coefficients between the physical experiments and the physical model (WFE) in our Bayesian inference approach. As the WFE model was replaced by the surrogate model, the output error now incorporates both the error between the surrogate model and the physical experiments, as well as the error between the surrogate model and the physical model. And the uncertainty between the surrogate model and the physical model is relatively small. However, this is a valuable point to give uncertainty of the surrogate model alone. We will investigate it in future work.

6) Metropolis Hastings algorithm is first introduced in the Figure 8. Apparently no introduction is present in the manuscript, while the authors say in Section 5 “Similarily to the previous case study, the inferred processes carried out using the M-H algorithm”

Response: The authors thank the reviewer for pointing out this issue. The introduction of M-H algorithm has been added. The details of the algorithm implementation have also been added to Appendix Section. Please check the details in lines 139-148, 253-254 and 407-415 of the revised manuscript.

7) Review of typos is needed

Response: The author has thoroughly checked the entire manuscript. Please note that this article uses British English conventions.

[R1] Sasena, Michael James. Flexibility and efficiency enhancements for constrained global design optimization with kriging approximations. University of Michigan, 2002.

Reviewer 2 Report

In this paper, a methodology to detect defects in joints is presented. The characterisation of the defect is investigated through a Bayesian inverse approach and a numerical approach based on finite element discretisation. To circumvent the computational issues due to the high calculation cost, the Bayesian procedure used a reduced surrogate of the numerical model to obtain the scattering coefficients. The method is presented and discussed for a particular case study corresponding to an aluminium joint with a circular hole as the defect. Experimental results are also presented. The quality of the manuscript is good both in terms of scientific content and communication. The paper shows some novelty in the application of a relatively new approach that can be furtherly investigated for practical application in the SHM. Therefore the reviewer recommends the manuscript publication after some minor revisions. These are listed in the "comments to author" sections.

Detailed comments.

1) The paper shows the application of the procedure to a specific case. This must be specified in the introduction since the application of the procedure to different cases could rise some implementation issues.

2) section 3. Ln 105. “rigorously quantify uncertainty”. The authors can remove “rigorously” and specify the type of uncertainties assumed in the model.

3) The joint considered in the paper has some special features. However, some real situations can show more complicated scattering behaviour due to non-linearities and mode conversions. Moreover, the frequency range and signal source can affect the results. The authors should introduce these aspects and cite the following paper: “Wave scattering from discontinuities related to corrosion-like damage in one-dimensional waveguides. Journal of The Brazilian Society of Mechanical Sciences and Engineering, 42 (10), 1-17, 2020”;

4) The method proposed in section 3.2.1, is also presented in “A finite element method for modelling waves in laminated structures, 2013, Advances in Structural Engineering, pp. 61-75”. For complete information, the authors must cite this paper.

5) Conclusions. Considering the previous comments, the authors should enrich the conclusions regarding the results presented in the paper while the description of future works should be removed.

Author Response

The authors would like to thank the reviewer for her/his constructive comments. We hope that we have adequately addressed the comments below and we are looking forward to receiving your response on our manuscript.

Reviewer #2: In this paper, a methodology to detect defects in joints is presented. The characterisation of the defect is investigated through a Bayesian inverse approach and a numerical approach based on finite element discretisation. To circumvent the computational issues due to the high calculation cost, the Bayesian procedure used a reduced surrogate of the numerical model to obtain the scattering coefficients. The method is presented and discussed for a particular case study corresponding to an aluminium joint with a circular hole as the defect. Experimental results are also presented. The quality of the manuscript is good both in terms of scientific content and communication. The paper shows some novelty in the application of a relatively new approach that can be furtherly investigated for practical application in the SHM. Therefore the reviewer recommends the manuscript publication after some minor revisions. These are listed in the "comments to author" sections.

Detailed comments.

1) The paper shows the application of the procedure to a specific case. This must be specified in the introduction since the application of the procedure to different cases could rise some implementation issues.

Response: The limitation of the proposed framework has been added in introduction and conclusion. Please check the details in lines 48-49 and 292-298.

2) section 3. Ln 105. “rigorously quantify uncertainty”. The authors can remove “rigorously” and specify the type of uncertainties assumed in the model.

Response: The authors thank the reviewer for this comment. It has been removed.

3) The joint considered in the paper has some special features. However, some real situations can show more complicated scattering behaviour due to non-linearities and mode conversions. Moreover, the frequency range and signal source can affect the results. The authors should introduce these aspects and cite the following paper: “Wave scattering from discontinuities related to corrosion-like damage in one-dimensional waveguides. Journal of The Brazilian Society of Mechanical Sciences and Engineering, 42 (10), 1-17, 2020”;

Response: The limitation mentioned in the above paper and the citation have been added to the revised manuscript. For more information, please check the response to Comment 1).

4) The method proposed in section 3.2.1, is also presented in “A finite element method for modelling waves in laminated structures, 2013, Advances in Structural Engineering, pp. 61-75”. For complete information, the authors must cite this paper.

Response: The citation has been added.

5) Conclusions. Considering the previous comments, the authors should enrich the conclusions regarding the results presented in the paper while the description of future works should be removed.

Response: The authors thank the reviewer for this comment. We have removed the description of future work related to joint application and added the limitation of the proposed framework.

Reviewer 3 Report

The article presents a Bayesian inference framework to identify the size of the circular hole in joints using frequency domain damage features. The article requires significant improvements in terms of the description of the methodologies used in the article. The connection between sections is missing. The authors are recommended to address the below queries before being considered for publication.

1) IN the abstract, it is described:  "Furthermore, a Kriging surrogate model is integrated with WFE to generate a database containing measured scattering properties and to replace the WFE as the forward model within probabilistic inference, which drastically enhances the computational efficiency of the proposed scheme."

The statement is confusing. Does Kriging surrogate model with WFe used as a predictive model? If it is the case, then how is it replacing the WFE model? Does it mean to say that the surrogate model was used in place of the WFE model? Please improve the clarity of the sentence and the entire abstract as well.

The statement is confusing. Does Kriging surrogate model with WFe used as a predictive model? If it is the case, then how is it replacing the WFE model? Does it mean to say that the surrogate model was used in place of the WFE model? Please improve the clarity of the sentence and the entire abstract as well.

2) On line 83, it is mentioned that: Sensor #1 is used to generate an input 84 waveform, and the rest of the PZT sensors receive the reflected and scattered signals. There is no sensor #1 labeled in figure 3 or figure 2. Please correct the sentence. 

3) It is difficult to see the difference between 3 mm and 4mm in figure 4. Please add a plot showing zoomed part of the time signal which shows the difference between different signals.

4) What are the model parameters? Is it defect (r) and standard deviation (sigma) of error? Please explicitly state it!

5) How does Eq.3 fit into the Bayes theorem? It looks like a likelihood function.

6) How does Eq. 4 arrive? What are the assumptions used for errors to arrive at this term? Are errors are assumed to iid?

7) The workflow is the heart of the entire article and gives an overview of the entire article. Only a single line is mentioned about the workflow. Please explain in detail about the workflow.  

8) What is the final posterior PDF? 

9) The posterior is expressed in Eq. (1) but the likelihood equation is not expressed anywhere in the article. How is the likelihood equation related to Eq. (3)? 

10) Is Eq.3 a likelihood term? if so, the variables used are mismatched from Eq.1. Please correct the equations and provide an explanation connecting all the equations defined in this article.

11) In the figure it is mentioned that the M-H sampling algorithm is used. Is it Metropolis–Hastings algorithm? Please use the complete name when using it for the first time. If M-H is Metropolis–Hastings, please mention that it is the Markov chain Monte Carlo sampling method. Please give references for the M-H algorithm and include a brief description of what M-H does and how it approximates the posterior PDF. In the figure, it is mentioned that M-H solves the posterior PDF. But M-H generates samples from the posterior to approximate the posterior.

The rest of the queries are posted as comments in the attached manuscript at the appropriate location.

Author Response

The authors would like to thank the reviewer for her/his constructive comments. We hope that we have adequately addressed the comments below and we are looking forward to receiving your response on our manuscript. Please check the attached revision letter and manuscript.

Round 2

Reviewer 3 Report

The authors have addressed most of the reviewer's comments but some of the modifications are incomplete. Please address the below address to be considered for publication.

1) For point 15, the MCMC chains are added to the author's response but not included in the revised manuscript.  The total sample number was specified as 100,000 and the burn-in period was specified as 20,000. However, none of them have been included in the revised manuscript. Please include.

2) Figure 5 is not clear. Could you please show the zoomed part of only one or two waveforms?

3) What is the time resolution of the signal? Please mention the sampling frequency of the signal.

The rest of the comments are included as comments on the revised pdf document.

Author Response

The authors would like to thank the reviewer for her/his constructive comments. We hope that we have adequately addressed the comments below and we are looking forward to receiving your response on our manuscript.

The authors have addressed most of the reviewer's comments but some of the modifications are incomplete. Please address the below address to be considered for publication.

1) For point 15, the MCMC chains are added to the author's response but not included in the revised manuscript.  The total sample number was specified as 100,000 and the burn-in period was specified as 20,000. However, none of them have been included in the revised manuscript. Please include.

Response: The total sample number and burn-in period have been added in the revised manuscript. Please check the details in lines 254-256. The image of the chains shows the random walk of the Markov chain, indicating that the Metropolis-Hastings algorithm is exploring the target distribution. In this study, we used so many samples (100,000) that the image of the chains was compressed, which provides little information to the reader. The author believes it would be best not to include it in the manuscript.

2) Figure 5 is not clear. Could you please show the zoomed part of only one or two waveforms?

Response: We apologise for the unclear display of the picture. The authors have inserted vector graphics, which allow for scalability without losing clarity. In this way, the difference between the signals can be clearly seen in the zoomed-in picture.

3) What is the time resolution of the signal? Please mention the sampling frequency of the signal.

Response: The time resolution of the signal is 1.0417e-07 second. The authors used a sampling frequency of 9.6 MHz, which has been added in the revised manuscript in line 97.

4) The labels are small and is not visible. Please increase the font size.

Response: The font size has been enlarged.

5) In the previous review, it is recommended to add more references related to adhesive joints but none as been cited added related to adhesive joints. The authors are recommended to cite this article: https://doi.org/10.2514/1.J060979

Response: The citation has been added in lines 24-25.